# FROM MINUTES TO DAYS: SCALING INTRACRANIAL SPEECH DECODING WITH SUPERVISED PRETRAINING

## ABSTRACT

Decoding speech from brain activity has typically relied on limited neural recordings collected during short and highly controlled experiments. Here, we introduce a framework to leverage week-long intracranial and audio recordings from patients undergoing clinical monitoring, effectively increasing the training dataset size by over two orders of magnitude. With this pretraining, our contrastive learning model substantially outperforms models trained solely on classic experimental data, with gains that scale log-linearly with dataset size. Analysis of the learned representations reveals that, while brain activity represents speech features, its global structure largely drifts across days, highlighting the need for models that explicitly account for cross-day variability. Overall, our approach opens a scalable path toward decoding and modeling brain representations in both real-life and controlled task settings.

## 1 INTRODUCTION

Modeling and decoding the neural representations of speech from brain activity has been a long-standing challenge in neuroscience (Mitchell et al., 2008; Mesgarani et al., 2014; Défossez et al., 2023; Tang et al., 2023). To address this problem, most studies rely on short (20 min - 1 hour) experiments in which participants listen to carefully-crafted auditory stimuli. Linguistic features (e.g. phonemes, words) can then be time-stamped and aligned with brain activity, allowing researchers to model or decode the neural representations of speech.

While successful, this approach remains highly constrained by the quantity and quality of brain recordings. For example, functional magnetic resonance imaging (fMRI) and electro- or magneto-encephalography (EEG/MEG) can be collected over several one-hour-long sessions (LeBel et al., 2023; d'Ascoli et al., 2024; Özdogan et al., 2025; Armeni et al., 2022). However, fMRI provides limited temporal resolution, while EEG and MEG offer limited spatial resolution. Conversely, intracranial recordings (iEEG) offer both high spatial and temporal resolution, but patients implanted with electrodes generally agree to participate in research experiments for only a few minutes at a time (Mesgarani et al., 2014; Zada et al., 2025; Herff et al., 2015). Overall, the brain activity used to model and decode speech perception is typically restricted to the moment where participants perform a specific cognitive task.

This task-focused approach likely under-utilizes the existing brain recordings. During iEEG implantation, patients with epilepsy typically spend about a week in a specialized monitoring unit, where continuous brain activity, video, and audio are recorded (Kim et al., 2020). This 24/7 clinical setup generates (1) more than 100X the amount of data typically analysed in research experiments, and (2) brain recordings that are paired with the visual and auditory environment of the patient. These large-scale neural and audio data are typically discarded, presumably because no well-established tool, framework, or model exists to analyze such uncontrolled recordings. Current approaches to enhancing decoding performance on limited iEEG task data typically include innovations on architecture (eg. Zheng et al. (2024)),

cross-subject or cross-session transfer learning (Singh et al., 2025; Wu et al., 2025; Memar et al., 2024), or self-supervised pretraining (Zhang et al., 2023; Chau et al., 2025; Yuan et al., 2024). However, there is no existing framework in which large-scale weeklong non-task neural and auditory recordings can be leveraged to improve decoding performance on downstream tasks.

Here, we frame this challenge as a supervised pretraining problem. The goal is to (1) maximally align week-long brain recordings with embeddings of the ambient environment sounds captured by the hospital room camera, and (2) evaluate whether this pretraining improves the modeling and decoding of the controlled experimental task. We focus on the auditory modality for two main reasons. First, the ambient audio captured by the camera closely reflects what the patient hears at each moment. However, the video stream, being allocentric, differs substantially from the patient's visual experience and is likely to be more challenging to align to brain activity. Second, recent work has shown that self-supervised audio models represent sounds in a way that is directly comparable to the brain's representations (Millet et al., 2022; Li et al., 2023; Vaidya et al., 2022; d'Ascoli et al., 2025).

Following a standard brain decoding architecture (Défossez et al., 2023), we adapt a contrastive learning approach (CLIP, Radford et al. (2021)) to align brain activity with representations from a pretrained speech model (wav2vec 2.0, Baevski et al. (2020)). We evaluate this approach on three patients implanted with stereotactic electrodes for one week, each of whom also completed an audiobook listening experiment (Evanson et al., 2025) lasting on average 120 minutes.

## 2 METHODS

### 2.1 PROBLEM FORMALIZATION

Brain decoding of speech can be formalized as predicting or retrieving a feature vector $V \in \mathbb{R}^d$ of a fixed-length audio segment $Y \in \mathbb{R}^T$ from the corresponding neural signals $X \in \mathbb{R}^{n \times t}$ with $t$ time steps from a $n$-channel recording. Formally, let $U = f(X)$ be the embedding of brain signals optimized for this goal, where $U \in \mathbb{R}^d$.

**Objective.** Contrastive learning has recently proved an efficient approach for decoding brain activity (Défossez et al., 2023; d'Ascoli et al., 2024). This approach consists of optimizing a CLIP objective (Radford et al., 2021) which maximizes the cosine similarity between positive pairs $(V_i, U_i)$ while minimizing it for negative pairs $(V_i, U_{i \neq j})$. Specifically:

$$\mathcal{L} = -\frac{1}{N} \sum_{i=1}^{N} \log \frac{\exp(t \cdot \cos\_sim(\mathbf{U}_i, \mathbf{V}_i))}{\sum_{j=1}^{N} \exp(t \cdot \cos\_sim(\mathbf{U}_i, \mathbf{V}_j))},$$

where $t = exp(t')$ and $t'$ is a learnable parameter.

**Goal.** However, this approach has previously been restricted to decoding the *true* speech sounds presented in a controlled experiment, e.g. an audio-book (Défossez et al., 2023; Özdogan et al., 2025; Jayalath et al., 2025a; Wang et al., 2023a). Instead, we aim to leverage the week-long recordings of iEEG paired with their *ambient* sounds to improve decoding of the controlled experiment.

### 2.2 APPROACH

**Architecture** To learn $f$, the transformation of brain recordings, we use the deep convolutional model of Défossez et al. (2023). The model takes in 3-second-long $n$-channel neural recording segments sampled at 40 Hz, where $n$ varies across subjects (Table 1), and outputs a vector with dimension $d$ that matches the audio feature vector. As we do not train our model across participants, we 1) remove the subject-specific layer of the model as we focus on within-subject decoding results, and 2) remove the spatial attention module that combines data from different neural channels based on their spatial location. The model has an initial linear layer that projects neural signals into a higher-dimensional space, a

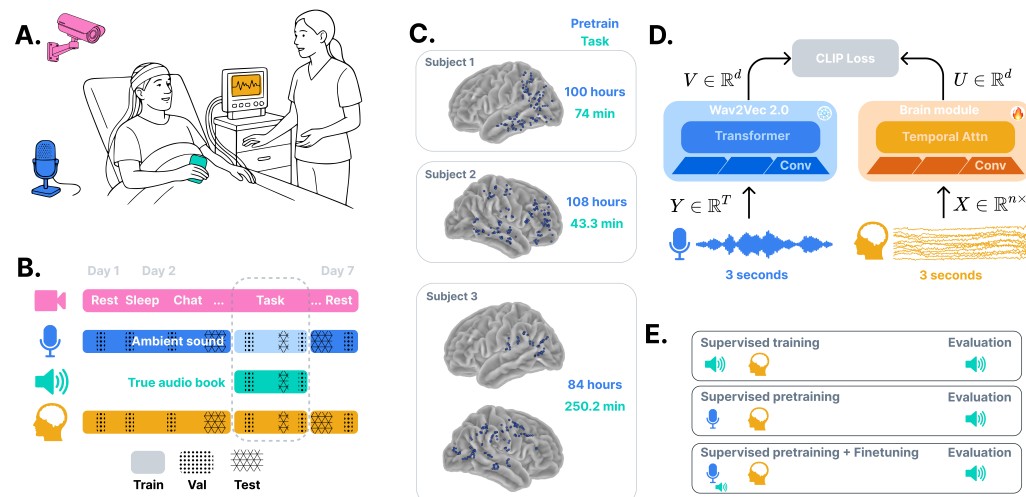

Figure 1: **Setup. A. Data collection**. Patients clinically implanted with intracranial electrodes spend a week in a monitoring unit where audio/video and brain recordings are recorded 24/7. During each patient's stay, they performed a controlled task consisting of listening to the audiobook of "The Little Prince" (de Saint-Exupéry, 1943) played on a smart phone. **B. Data structure.** Audio/video and brain recordings are dense time series. During the task (grey dashed box), we also know the true sound of the audiobook (green). Pretraining and validation are based on data outside the audiobook task period. Finetuning is based on neural and audio data captured during the task. Except if stated otherwise, evaluation is based on held-out task data from the true audiobook data. **C. Electrode localization and pretraining vs task data quantity per participant.** Electrodes are projected to the closest cortical surface for clarity. **D. Model**. We use contrastive learning between a pretrained sound module (Baevski et al., 2020) and a brain module (Défossez et al., 2023). **E. Approaches**. We compare three main approaches. All are evaluated on held-out data from the same audiobook task. (1) A baseline approach where the model is trained to align brain activity to the true audiobook sounds. (2) A zero-shot approach where a model is pretrained to align brain activity to ambient sound. (3) A pretraining + finetuning condition where the pretrained model is finetuned on the true audiobook sounds.

stack of convolutional blocks with skip connections, and a Bahdanau attention layer (Bahdanau et al., 2015) at the end of the temporal convolutions to aggregate over the temporal dimension.

**Audio preprocessing.** Similarly to Défossez et al. (2023), we use a pretrained wav2vec 2.0 model to generate $V$, i.e. the latent representation of an audio segment. We run the wav2vec2-large-xlsr-53 [1] model in inference mode and obtain the activations of the 19th layer, as a previous study showed that deeper layers in the model map better linearly to the brain (Millet et al., 2022). For this, we split sound recordings into 30-second chunks and discarded chunks shorter than 10 seconds at the end of each 2-hour continuous recording. We obtain model embeddings from these 30-second audio chunks, and interpolate the embeddings from 50 Hz to 120 Hz. We then extract 3-s segments using non-overlapping sliding windows and average the latent activations across token positions to obtain one vector per segment. During pretraining or finetuning, we apply Z-scoring of the wav2vec 2.0 features with the statistics of the train split of the respective datasets. In the zero-shot evaluation, the statistics of the train split of the pretraining dataset are applied across evaluations on different datasets.

---

[1]https://huggingface.co/facebook/wav2vec2-large-xlsr-53

**Neural preprocessing.** The iEEG data was first band-pass filtered between 0.05-50 Hz then downsampled to 40 Hz with MNE-Python (Gramfort et al., 2013). Each channel was then independently normalized using a robust scaler from scikit-learn (Pedregosa et al., 2011).

## 2.3 DATA

**Participants and ethics.** The study includes three subjects who underwent iEEG recordings as part of their treatment for intractable epilepsy. They participated in Evanson et al. (2025) and their week-long data during hospitalization were saved. The study was approved by the National Ethics Committee and the Local IRB. The participants and, when applicable their legal guardians, provided informed consent. Participation was voluntary, had no impact on participants' clinical care, and did not include any form of compensation. All research data is stored securely at the Hospital and was processed exclusively by its staff.

**Brain, video, and audio recordings.** The neural data were recorded from stereotactic EEG. The ambient sound was extracted from the video recordings generated by the clinical recording system. These recordings run 24/7 for the duration of each participant's hospitalization.

### 2.3.1 DATASETS

We split the prepared dataset into a large pretraining dataset and two smaller, downstream datasets for finetuning and testing.

**Downstream speech dataset.** During their stay, participants agreed to listen to the "The Little Prince" audiobook (de Saint-Exupéry, 1943). We use the original sound files of the audiobook and the corresponding brain activity for our downstream dataset (Table 1) – hereafter referred to as "true audiobook sounds". We also use the (noisy) ambient sound recorded from the video system during this task – hereafter "ambient audiobook sounds".

**Pretraining week-long dataset.** The pretraining dataset consists of the ambient sound and the corresponding brain signals during daylight hours between 6:00 and 23:00 recorded during the participant's hospital stay. This part of the audio data is hereafter referred to as "ambient sounds". The data when participants are listening to the audiobook is excluded from this pretraining set (Figure 1). The amount of data used for pretraining per participant is described in Table 1.

**Train, validation, and test splits.** For the week-long pretraining dataset, recordings are randomly split into train (90%), validation (5%), and test (5%) by recording files, each 2 hours long. For each of the task evaluation datasets, we randomly split 30-second chunks of neural and audio recordings into train (80%), validation (10%), or test (10%) sets. After splitting, the recordings are further segmented into 3-second non-overlapping clips for model training. This process is performed for each participant independently.

## 2.4 EXPERIMENTS.

**Optimization.** We trained our model per subject using AdamW (Loshchilov & Hutter, 2019) optimizer with a learning rate of $10^{-4}$ and a batch size of 128. We used the One Cycle Learning rate scheduler (`max_lr=2e-4` and `pct_start=0.3`) with a linear annealing strategy (Smith & Topin, 2018), which gradually increases and eventually decreases the learning rate over training. The model was trained for 100 epochs, or until the median relative retrieval rank on the validation set had not improved in the last 10 epochs. The same hyperparameters are used for pretraining and finetuning.

**Evaluation.** Our baseline approach consists of training a model exclusively on the true audiobook sounds, consistent with typical cognitive decoding approaches (Défossez et al., 2023). Our main approach consists of pretraining a model on the week-long ambient sounds,

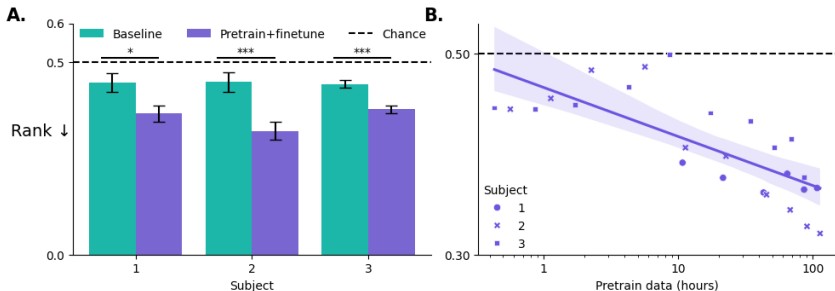

Figure 2: **Pretraining improves performance on downstream task.** **A.** Mean retrieval rank on true audiobook test set. Baseline method compared to pretraining+finetuning. Error bars indicate standard error of the mean (SEM) across samples in the respective test sets. Stars denote statistically significant differences in mean retrieval rank between test sets. **B.** True audio test set performance scales with increasing hours of pretraining data. A log-linear regression line is fitted, with shading indicating 95% confidence interval.

and finetuning it on the true audiobook. We add a linear head to the pretrained model during finetuning. Model performance is evaluated using the mean relative retrieval rank. This metric quantifies the relative rank of the true audio segment within a retrieval set. A relative rank of 0 (or 1) indicates that the predicted sound feature $U$ is the most (or least) similar to the true sound feature $V$ out of the retrieval set. To test whether performance varies within subjects between conditions, we use the non-parametric two-tailed Mann-Whitney U test.

**Representational analyses** To help interpret the representations of the brain signals captured by our model, we apply UMAP dimensionality reduction (McInnes et al., 2020; Sainburg et al., 2021) to the wav2vec 2.0 embeddings and brain module embeddings and explore four features: the recording date of the iEEG (from day 1 to day 7), the time of the recording (from 6:00 to 23:00, for an analysis of inclusion of night time data see Figure 11), the melspectrum centroid, and whether the segment contains speech or not using Whisper (Radford et al., 2023). We use hyperparameters `n_neighbors=50`, `min_dist=0.8` for the UMAP algorithm and fit using the cosine distance with 2 components.

We also conduct a linear decoding analysis of these feature spaces to quantify the extent to which these features are decodable from the embedding space. We fit a ridge regression model (RidgeCV) from scikit-learn (Pedregosa et al., 2011) with 7 regularization values log-linearly spaced between $10^{-3}$ and $10^3$ (`alpha_per_target` = True). The embeddings are standardized to have zero mean and unit variance before model fitting. We use a group k-fold (k=5) cross-validation where each group is the 30-second chunk the embedding belongs to. Each group is split either in the train (80%) or test (20%) set. The Pearson correlation is measured between the predicted and true feature values.

## 3 RESULTS

### 3.1 PRETRAINING ON WEEK-LONG DATA IMPROVES SPEECH DECODING PERFORMANCE

**Supervised pretraining improves downstream task performance.** We first evaluate whether there is a performance gain from pretraining the model on week-long data. Relative retrieval ranks are measured for two models on the test split of the true audiobook dataset: a pretrained model fine-tuned on the training split, and a baseline model trained from scratch on the same split. For all three subjects, the pretrained model significantly outperforms the baseline model (Subject 1: p-value=0.017, Subject 2: p-value<10e-3, Subject 3: p-value<10e-3), demonstrating the benefit of supervised pretraining for downstream tasks (Figure 2A). Interestingly, we find that finetuning a pretrained self-supervised model does not reach the same level of performance as the supervised pretrained models (Figure 15).

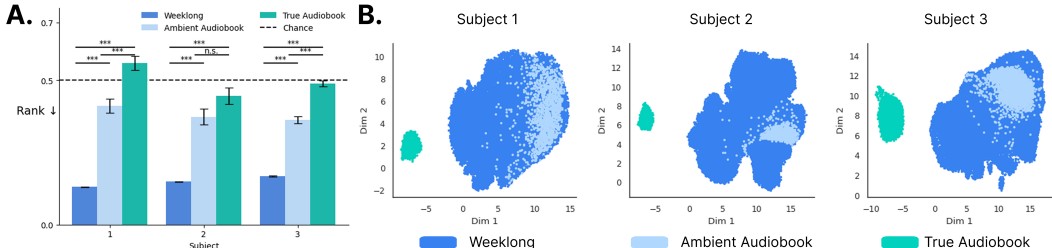

Figure 3: **The true audiobook sounds is out of distribution from the week-long ambient sounds.** **A.** Zero-shot mean retrieval rank on test sets of the pretraining dataset, the ambient audiobook dataset, and true audiobook dataset. Error bars indicate SEM across samples in the respective test sets. Stars denote statistically significant differences between test sets. **B.** Umap clustering of wav2vec embeddings colored by dataset for Subject 1, Subject 2, and Subject 3.

**Log-linear scaling of task performance with pretraining data quantity.** Does more pretraining data lead to better test set performance? To answer this question, we apply the same evaluation on models pretrained with different data quantities sampled at 10%, 20%, 40%, 60%, 80%, and 100% for each subject. We find that retrieval rank decreases log-linearly with increasing hours of pretraining data (Figure 2B). Furthermore, performance does not appear to plateau even in the largest data regimes, suggesting that more pretraining data could further improve performance.

### 3.2 FINETUNING PRETRAINED MODELS TO ALIGN TO TASK DATA

**Zero-shot performance is poor in generalization conditions.** We then ask how generalizable the features learned during pretraining are for different testing conditions. We therefore assessed the zero-shot retrieval performance of the pretrained model on the test split of three datasets: week-long pretraining data, ambient audiobook data, and true audiobook data. The mean retrieval ranks for the two downstream datasets (ambient=0.382, true=0.498) are significantly higher than those of the week-long data (rank=0.149) (Figure 3A). Without finetuning, the pretrained model does not directly generalize to task data.

**Target data distribution shift.** We hypothesize that this could be due to a distribution shift between the datasets. The week-long data and the ambient audio data are both captured by a microphone from the recording system in the room, yet the sound quality of typical human speech or other ambient sounds differs to that of the audiobook played through the phone speaker during the task. The true audiobook data is even more different as it contains the clean audio files that does not include any background sounds that might arise during the task.

To confirm the degree of distribution shift from the pretraining dataset to the ambient and true audiobook data, we apply UMAP clustering of the wav2vec feature vectors of audio segments with the following data from each subject: 20% randomly sampled pretraining data segments, all ambient audiobook segments, and all true audiobook segments (Figure 3B). We observe that while the ambient audiobook segments reside in a subspace of the overall pretraining distribution, the distribution of true audiobook data is drastically different from the rest. This confirms that the true audiobook labels for the speech decoding task is indeed out of distribution (OOD) to the pretraining data.

**Adapt to distribution shift with finetuning.** The OOD nature of the true audiobook data likely explains the poor zero-shot performance of the pretrained model despite the performance gain we observe with pretraining (Figure 4B, Figure 2A). We therefore hypothesize that the model is increasingly adapting to the distribution shift during finetuning. To test this, we finetune a pretrained model on varying amounts of true audiobook data sampled at the same ratio as previously described and assess their performance on the true audiobook

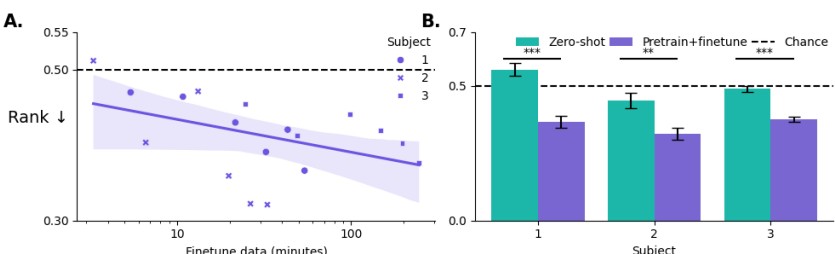

Figure 4: **Finetuning on greater quantities of task data improves decoding performance.** **A.** Mean retrieval rank over increasing minutes of finetuning data. The curve represents a log-linear fit to the data, and the shading indicates the 95% confidence interval across all data points. **B.** Finetune improves performance over zero-shot condition of the true audiobook on the pretrained model. Error bars indicate SEM across samples in the respective test sets. Stars denote statistically significant differences between test sets. Zero-shot metrics from Figure 3A and pretraining+finetuned metrics from Figure 2A are replotted here for a clear comparison.

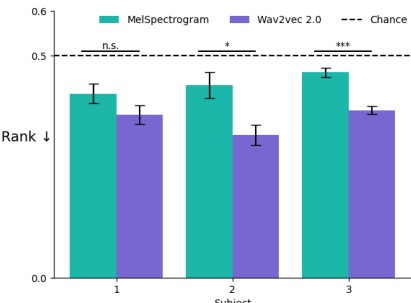

Figure 5: **Deep learning features offer advantages in neural speech decoding over classic auditory features.** Pretraining and then finetuning on the true audiobook, using either melspectrogram or wav2vec 2.0 features. Error bars indicate SEM across samples in the respective test sets. Stars denote statistically significant differences between test sets.

test set. A similar log-linear trend is observed where increasing minutes of finetuning data improves performance (Figure 4A). These results highlight that despite the poor direct generalization of the pretrained model, the features learned during pretraining are transferable with finetuning as the model adapts to the distribution shift (Figure 4B).

### 3.3 THE IMPACT OF CONTEXTUALIZATION OF THE SOUND FEATURES

Next, we ask whether using contextualized sound features from wav2vec 2.0 offer any advantage over more classic audio features. We therefore pretrain and then finetune the brain module on melspectrogram (see Section 4.1 for feature extraction details) and compare its performance to the model previous presented (pretrained and finetuned on wav2vec features). We find that models trained with wav2vec features significantly outperform those trained with melspectrogram for two of the three subjects (Figure 5, Subject 1: p-value=0.132, Subject 2: p-value=0.010, Subject 3: p-value<10e-3), indicating that the brain module aligns better to a contextualized embedding space. This difference across subjects could be explained by the variance in the electrode localization (Figure 1C). The electrodes of Subject 1 are located closer to the primary auditory cortex in the left hemisphere, which is known to contain lower-level acoustic information (Price, 2010; Mesgarani et al., 2014; Huth et al., 2016), than the electrodes of the other two subjects which are scattered across the cortex. Interestingly, the impact of contextualization is not significantly affected by the duration of context (Figure 10).

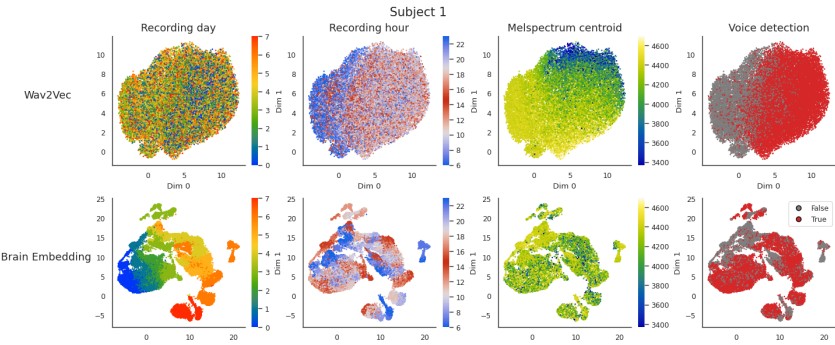

Figure 6: **Learned embeddings track the drift in neural signals across days.** Umap clustering of wav2vec (top row) and brain embeddings (bottom row) colored by recording date, recording hour, melspectrum centroid, and speech. The computation of melspectrum centroid is detailed in Section 4.1. Only segments included in the pretraining set are plotted (day-time only). UMAPs for the other subjects are included in Section 4.8.

### 3.4 Interpreting the model embedding space

To investigate the representations learned by the pretrained model, we apply UMAP dimensionality reduction to both the wav2vec and brain embedding spaces on a randomly sampled 20% subset of week-long data from Subject 1. The resulting 2D embeddings were visualized using four distinct color maps: recording date and recording hour, melspectrum centroid (reflecting the center of mass of the melspectrogram), and a voice detection label.

The wav2vec embedding space does not exhibit distinct clustering based on recording dates. However, it shows a coarse organization into two main clusters. One cluster consists of recordings without speech and mostly from the early morning and late evening, a period likely corresponding to patient rest. For lower-level acoustic features, the melspectrum centroid is well-represented within the wav2vec embeddings. In contrast, the brain embedding space exhibited significant and distinct clustering based on the recording date, likely due to the distribution shift in neural signals across days. Sub-clusters seem to form within each date-specific cluster, separating segments from different recording hours and segments containing speech or not. Unlike the wav2vec embeddings, the Melspectrum centroid was not globally reflected in the brain embedding as a smooth gradient.

We further quantify this effect by assessing the linear decodability of these features from the wav2vec and brain module embedding spaces with a ridge regression model (Figure 14). We compute the Pearson correlation between the true and predicted feature values on each of the cross-validation test splits for each subject. Context features such as recording date is more linearly decodable ($r = 0.95 \pm 0.01$, mean $\pm$ SEM across subjects) from brain embeddings than from wav2vec ($r = 0.64 \pm 0.03$), while the reverse is observed for audio features including mel spectral centroid ($r = 0.44 \pm 0.02$ in brain embedding, $r = 0.98 \pm 0.00$ in wav2vec) and voice detection ($r = 0.62 \pm 0.03$ in brain embeddings and $r = 0.76 \pm 0.04$ in wav2vec, Figure 14). These results show that while the brain module learns to extract global wav2vec features to some extent, it still retains some modality-specific structure in the embedding space.

## 4 Discussion

**Contributions.** These results highlight three main contributions. First, we show that week-long recordings need not be discarded and can be leveraged through contrastive learning to improve the decoding of speech from brain responses recorded during a classic controlled experiment. Second, this approach effectively scales: the gain of our model performance increases log-linearly with the amount of pretraining data. Finally, this approach

reveals that, while iEEG representations contain rich speech features, their global structure varies largely across days.

**Scaling brain modeling.** To our knowledge, this work is the first to demonstrate a scalable method to improve speech decoding with intracranial recordings. This approach complements growing efforts to scale the decoding and modeling of iEEG (Berezutskaya et al., 2023; Wang et al., 2023a; Yuan et al., 2024; Zhang et al., 2023; Peterson et al., 2022; Memar et al., 2024), EEG (Wang et al., 2025; Jiang et al., 2024; Kostas et al., 2021), MEG (Jayalath et al., 2025b; d'Ascoli et al., 2024) and fMRI (Allen et al., 2022; Tang et al., 2023; Millet et al., 2022; Antonello et al., 2023). However these past efforts remained limited to one of three possible approaches. The first was based on linear modeling (e.g. Goldstein et al. (2025)), and thus scales poorly. The second is based on supervised architectures applied to task-only data (Willett et al., 2023; Metzger et al., 2023; Herff et al., 2015; Card et al., 2024; Wang et al., 2023b; Chen et al., 2024; d'Ascoli et al., 2024; Antonello et al., 2023), and is thus necessarily limited by the amount of brain recordings. The third approach is based on unsupervised models of brain recordings (Banville et al., 2021; Kostas et al., 2021). For example, Wang et al. (2023a) and Chau et al. (2025) proposed a self-supervised framework that trains solely on iEEG, but does not utilize paired audio and video data for extracting task-relevant representations. Our approach overcomes these limitations by aligning neural activity with continuous, naturalistic sensory inputs on far longer timescales than traditional task-based datasets allow. Our cross-modal supervision both grounds the neural representations in meaningful environmental structure and provides a scalable path toward models of brain activity that directly relate to cognition.

**Handling distribution shift between ambient and task data.** A significant challenge of our approach is the distribution shift between the week-long data and the task data – both in terms of how the brain may be activating, but also in how the true sound of the audiobook differs from the sounds captured by the ambient recording system. Here, we show that finetuning remains a necessary step to successfully address this issue. Note that zero-shot decoding performance on ambient audiobook sounds is significantly better than chance, suggesting that it is primarily the distribution shift of the audio recordings, rather than the task, that necessitates finetuning.

**Representation analysis.** Interpreting the activations of the brain (or of AI models) can provide insight into the nature of the underlying representations. In our analysis, we find that despite optimizing for the CLIP objective, the embedding of brain activity aligns in a non-trivial way with wav2vec 2.0 (Figure 3). Specifically, recording date and time are better captured by the brain embedding, whereas classic auditory features such as the Melspectrum centroid and the presence of speech are better captured by wav2vec 2.0 (Figure 14). This unexpected phenomenon suggests that brain activity varies substantially across the week. While further investigation is needed, this may be caused by the progressive reduction of seizure-reducing drug dosage over the course of hospital stay, which may in turn exacerbate pathological brain activity. Regardless of the underlying mechanism, this non-stationarity highlights the potential of designing models that are more explicitly robust under distributional drift. Methods such as domain-adversarial training (Purushotham et al., 2017), sliding-window normalization (Tanaka et al., 2022), or time-aware positional embeddings (Zhou et al., 2021) offer valuable future research directions for more drift-robust neural decoding models.

**Scaling further.** This work demonstrates how to effectively leverage nearly 100x more data than what is typically used for modeling and decoding speech from iEEG recordings. Importantly, the lack of scaling plateau suggests that even larger-scale – potentially multi-subject – datasets could further improve performance. This, however, will require solving the heterogeneity in electrode implantation across patients, for example with a subject-embedding layer (Défossez et al., 2023; Benchetrit et al., 2023; Chen et al., 2025; Careil et al., 2025).

**Multimodal alignment.** While we focused on the auditory modality, other modalities may provide complementary insights. Video data can capture posture, movement (Singh et al., 2020; Peterson et al., 2022), and social interaction, while alignment with text-based language models could enhance semantic representations—though this is currently limited by the absence of time-stamped transcripts. While this will likely require substantial effort to detect and time stamp individual words and body movement, extending this approach to integrate audio, text, and video embeddings is essential for extending the brain modeling of cognition. Overall, our results demonstrate the potential and scalability of this approach in improving neural speech decoding and modeling with real-life and controlled task data.

## Ethics Statement

The code of ethics for this experiment is described in the **Participants and ethics** in the Data section of the paper. We refer the readers to the original data paper Evanson et al. (2025) for further details on ethics approval.

## Reproducibility Statement

The preprocessing steps taken to prepare the datasets have been described in the Method section of the main text. The code for the model architecture by Défossez et al. (2023) is accessible here `https://github.com/facebookresearch/brainmagick`.

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

# Appendix

## 4.1 Melspectrogram feature extraction

We generated the melspectrogram of audio segments using 40 frequency bands, fft window size of 512, and a hop length of 128, and then transformed to a logarithmic scale for each 30 second audio chunk. The spectrogram is then resampled to 120 Hz and the power per time step within each 3-second audio segment window is averaged to get one vector for each time window. We also use this to compute the melspectrum centroid for the subsequent clustering analysis. The melspetrum centroid is computed using the librosa python library (McFee et al., 2023). For the UMAP of the melspectrum centroid, we plot points within the 1st and 99th percentiles to remove extreme outliers for visualization purposes.

## 4.2 Dataset description

Table 1: Description of pretraining and target task dataset per participant.

| Participant | # Channels | Pretraining (hrs) | LPP Data (minutes) |
|---|---|---|---|
| 1 | 141 | 100.44 | 74.0 |
| 2 | 214 | 108.36 | 43.3 |
| 3 | 230 | 83.80 | 250.2 |

## 4.3 Additional Controls

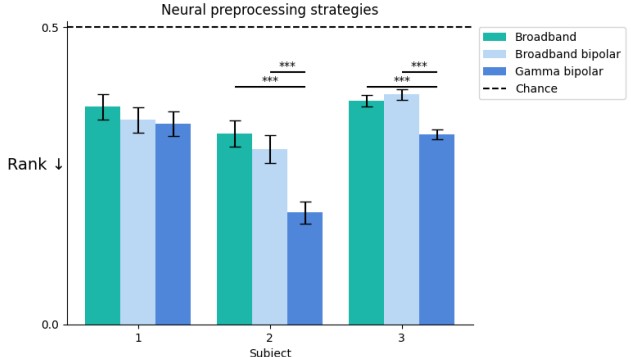

Figure 7: **A control for different neural preprocessing strategies.** We present results for 3 different neural preprocessing strategies. Broadband is the most simple and used throughout the paper. We compare that to Broadband bipolar and observed no statistical difference. Finally we compare to Gamma bipolar, for which we compute the gamma power on the bipolar constructs by filtering between (70, 120) Hz, and applying a Hilbert transform. We observe improvements in performance for subjects 2 and 3. We present the rank on the Pretrain+finetune setup.

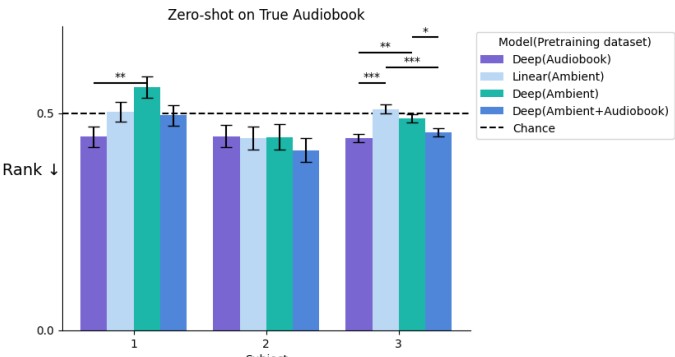

Figure 8: **A control for different pretraining strategies (evaluated in zero shot condition).** We present results for pretraining either the convolutional architecture presented in the rest of the paper (here labeled "Deep") or a linear layer. We compare pretraining on the true audiobook only, the ambient audio, or the ambient audio *and* the true audiobook. Note that Deep(Audiobook) is the same as "Baseline" and Deep(Ambient) the same as "Zero-shot" in the main text.

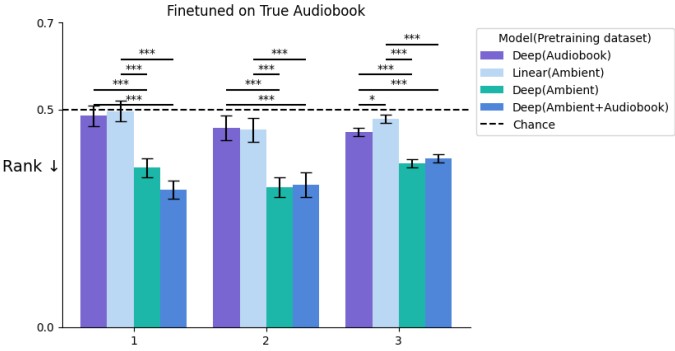

Figure 9: **A control for different pretraining strategies (evaluated after finetuning on the true audiobook).** We present results for pretraining either the convolutional architecture presented in the rest of the paper (here labeled "Deep") or a linear layer. We compare pretraining on the true audiobook only, the ambient audio, or the ambient audio *and* the true audiobook. Note that Deep(Ambient) is the same as "Pretrain+finetune" in the main text.

## 4.4 Impact of context duration of wav2vec

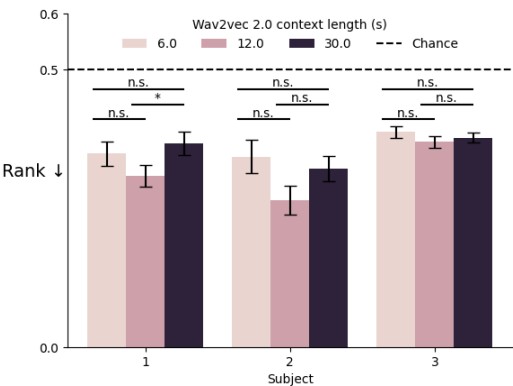

Figure 10: **The context length of wav2vec 2.0 has no clear impact on downstream decoding performance.** Pretraining and then finetuning on the true audiobook, using different wav2vec 2.0 context lengths. Error bars indicate SEM across samples in the respective test sets. Stars denote statistically significant differences between test sets.

## 4.5 Including night time data in pretraining

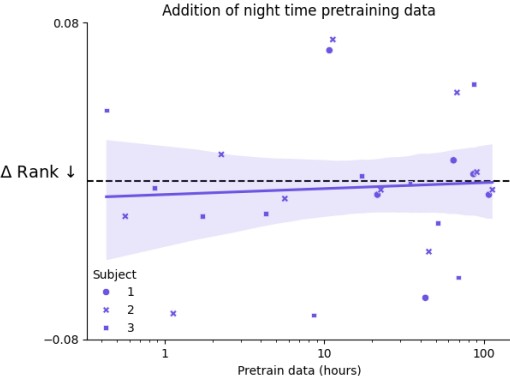

Figure 11: **There is no clear advantage in including the weeklong data from night time (23:00-6:00) in the pretraining set.** $\Delta$Rank = Rank(daytime pretraining + finetuning) - Rank(daytime+night time pretraining + finetuning), where finetuning is on the true audiobook. The curve represents a log-linear fit to the data, and the shading indicates the 95% confidence interval across all data points.

## 4.6 Decoding per Region of Interest in the Brain

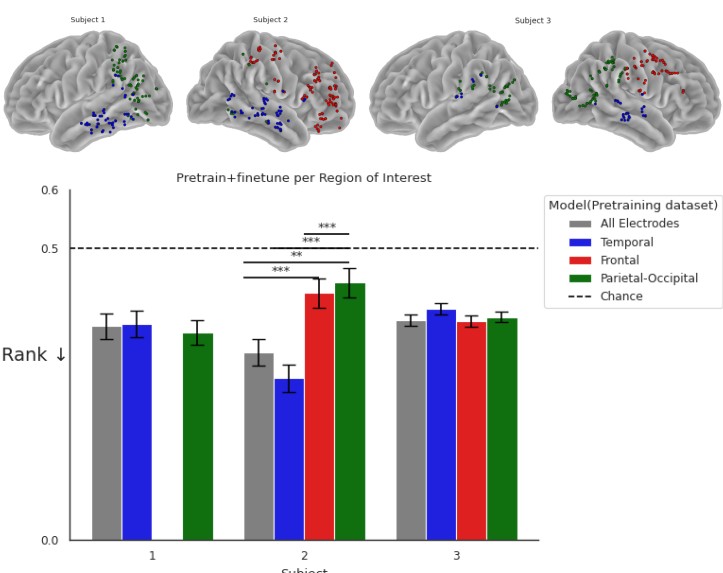

Figure 12: **Decoding per ROI**. We defined three regions of interest (ROI): temporal cortex, frontal cortex and the parietal-occipital cortices. We train and test a model using only electrodes from a given ROI. For the subject (2) with a difference in performance between ROIs we observe that the temporal cortex offers a performance improvement, while the frontal and parietal-occipital result in worse performance, which fits with the established role of the temporal cortex in auditory processing (Price, 2010; Fedorenko et al., 2024)

## 4.7 Voice Detection with Whisper

We use the `openai/whisper-large-v3` model from HuggingFace [2] and use the recommended configuration for speech detection in the original paper of the model (Radford et al., 2023). If no actual speech tokens are output from model inference on a 30-second speech segment, then the segment is labeled as no voice.

---

[2]https://huggingface.co/openai/whisper-large-v3

## 4.8  UMAP CLUSTERING FOR ALL SUBJECTS

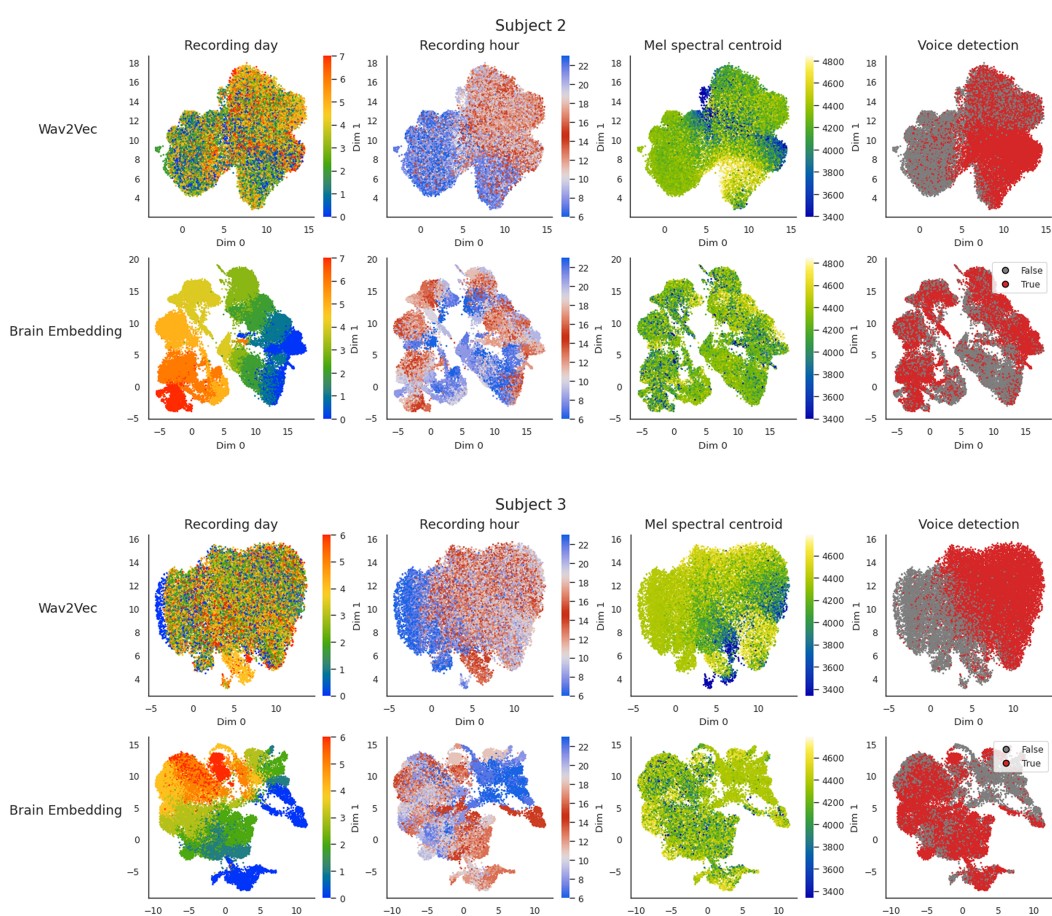

Figure 13: **UMAP clustering of Wav2Vec features like in Figure 2 of Subject 2 and Subject 3**. Data sampled similarly as Subject 1 as described in the main text.

## 4.9 Ridge results

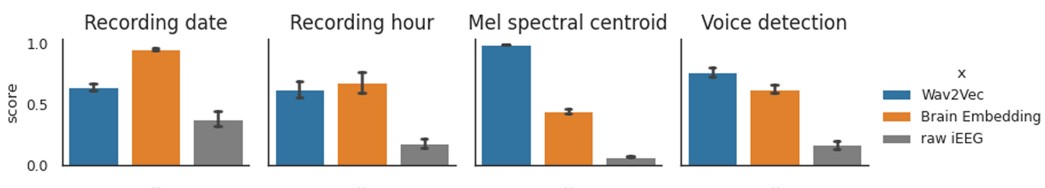

Figure 14: **Recording day and hour are more decodable from brain embeddnigs than from wav2vec.** Linear decoding scores for four different features across different embedding spaces including the raw iEEG. Bar plot shows the mean and SEM of Pearson correlation across subjects. For recording hour, we transform the hour labels to the absolute difference between the hour and noon (12:00) before fitting the ridge model.

## 4.10 Finetuning SSL model

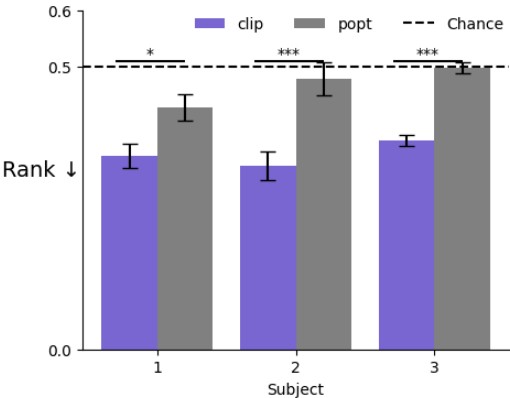

Figure 15: **Finetuning supervised vs self-supervised pretrained models.** We compare our pretrained model to a publicly available self-supervised (SSL) pretrained model Population Transformer (PopT) (Chau et al., 2025) on our downstream task. We take the pretrained PopT [3] finetuned with a linear layer that projects from the [CLS] token of the model to the dimension size of Wav2Vec embedding ($d = 1024$). We unfreeze the PopT weights and finetune with a learning rate of $5e - 5$ with all other hyperparameters kept the same. To use BrainBERT(Wang et al., 2023a), we preprocess the sEEG signal with a notch filter at $50\,\mathrm{Hz}$ and its harmonics, a highpass filter of $0.1\,\mathrm{Hz}$, and laplacian rereference. For each 3-second window, we apply the Short-Time Fourier Transform function provided by the authors and run BrainBERT in inference mode to obtain embeddings for each channel. The per-channel embeddings are then averaged along the temporal dimension and passed as input to PopT. The mean retrieval rank on the downstream task test set shows that our supervised pretraining models outperformed the finetuned SSL model.

## 4.11 LLM usage

After writing the initial draft, the authors have used LLMs to edit and polish parts of the main text.

