# OpenReview forum: "From Minutes to Days: Scaling Intracranial Speech Decoding with Supervised Pretraining"
_ICLR.cc/2026/Conference — Submitted to ICLR 2026_

### Official Review · Reviewer_awbe · 2025-10-27

**Soundness:** 2
**Presentation:** 2
**Contribution:** 1
**Rating:** 2
**Confidence:** 4

**Summary:**

This paper presents a framework to scale intracranial (iEEG) speech decoding by leveraging week-long, continuous brain and audio recordings as supervised pretraining data. Using a contrastive learning model that aligns brain activity with pretrained audio representations (wav2vec 2.0), the approach demonstrates significant gains over models trained solely on short, controlled datasets. The study reveals that pretraining performance improves log-linearly with the amount of data and that downstream performance on controlled tasks benefits robustly from large-scale pretraining followed by supervised finetuning. Analysis of the learned embedding spaces highlights issues of cross-day neural drift and distributional shift between ambient and experimental audio.

**Strengths:**

1. The methodology is clearly formalized, with careful and transparent documentation of preprocessing, architecture, and experimental protocols.

2. The empirical evaluation is rigorous: the impact of pretraining is shown clearly in Figure 2, and the log-linear relationship between data amount and downstream performance is rare in the brain decoding literature.

3. The analysis of representation drift (Figure 6 and 9) is a valuable, often neglected aspect, revealing new neuroscientific challenges that arise with longer time windows.

**Weaknesses:**

1. Both the model architecture and training paradigm are directly adopted from [1] (Line 133). The dataset was not collected by the authors, yet directly selected 3 subjects from 46 subjects in [2], which is not publicly available. The code link doesn’t belong to this project, but was copied from [1].

2. The experimental evaluation included only 3 subjects from [1] and lacked comparison with advanced sEEG decoding baselines [3-6], which makes it difficult to position the contribution of this article.

3. Although the idea of using sEEG-audio signal pairs during the non-task phase to improve decoding performance during the task phase is interesting, the experimental design itself is to ensure that the subjects focus on carefully designed cognitive tasks and that the recorded sEEG signals contain information about language perception, which makes the neuroscientific basis and reproducibility of this work questionable.

**References**:

[1] Défossez A, Caucheteux C, Rapin J, et al. Decoding speech perception from non-invasive brain recordings[J]. Nature Machine Intelligence, 2023, 5(10): 1097-1107.

[2] Linnea Evanson, Christine Bulteau, Mathilde Chipaux, Georg Dorfm¨uller, Sarah FerrandSorbets, Emmanuel Raffo, Sarah Rosenberg, Pierre Bourdillon, and Jean-R´emi King. Emergence of Language in the Developing Brain. Manuscript Online, May 2025. URL
https://ai.meta.com/research/publications/emergence-of-language-in-the-developing-brain/. (Accessed 10/09/2025).

[3] Wang C, Subramaniam V, Yaari A U, et al. BrainBERT: Self-supervised representation learning for intracranial recordings[J]. arXiv preprint arXiv:2302.14367, 2023.

[4] Chau G, Wang C, Talukder S, et al. Population transformer: Learning population-level representations of neural activity[J]. ArXiv, 2025: arXiv: 2406.03044 v4.

[5] Zhang D, Yuan Z, Yang Y, et al. Brant: Foundation model for intracranial neural signal[J]. Advances in Neural Information Processing Systems, 2023, 36: 26304-26321.

[6] Yuan Z, Shen F, Li M, et al. Brainwave: A brain signal foundation model for clinical applications[J]. arXiv preprint arXiv:2402.10251, 2024.

**Questions:**

See the above weaknesses.

---

> ### Author Response · Authors · 2025-11-27
> **Reply 1/3**
>
> We kindly thank the reviewer for recognizing the strengths of this paper. We hereby provide a point-by-point response to the concerns raised by the reviewers
>
> 1. _“Both the model architecture and training paradigm are directly adopted from [1] (Line 133). … The code link doesn’t belong to this project, but was copied from [1].”_
>
> The model architecture and loss function are of [1] (L102), and the code link provided explicitly states it is from [1] (L550), not our own.
>
> Our main contribution is not a novel architecture, but a new training approach that systematically leverages a large amount of previously discarded, out-of-distribution labeled data. We adopted a well-tested architecture from the literature to validate this novel training paradigm. No prior work has demonstrated how to utilize this specific type of large-scale labeled, but out-of-distribution, data for the downstream task.
>
> _“The dataset was not collected by the authors, yet directly selected 3 subjects from 46 subjects in [2], which is not publicly available”_
>
> We recorded 3 patients over 1 week each. Each of these patients listened to the audiobook. These brain responses to the audiobook have been reported in detail in Evanson et al. Such week-long recording is not available for other datasets. They are either not publicly accessible or have been discarded due to the lack of a clear framework to use the paired label together with the data.

---

> ### Author Response · Authors · 2025-11-27
> **Reply 2/3**
>
> 2. _“The experimental evaluation included only 3 subjects from [1]”_
>
> The data in [1] includes 46 patients who listened to the Little Prince audio but not all have weeklong recordings as for most of them weeklong data was discarded after clinical routines. Therefore, we are unable to include all the subjects from the original paper in this analysis.
>
> _“lacked comparison with advanced sEEG decoding baselines [3-6], which makes it difficult to position the contribution of this article.”_
>
> We appreciate the reviewer's request for comparison with self-supervised learning (SSL) sEEG decoding baselines [3-6]. We provide a new analysis demonstrating that our approach outperforms a state-of-the-art decoding baseline.
> - Model Availability: The codebase for [6] is not publicly available [link](https://github.com/yzz673/Brant-2). Since the authors of [4] demonstrated that PopT [3] outperforms [5] on iEEG decoding tasks, we focused our new experiments on PopT (a follow-up work to [3]).
> - New Experiment (PopT Comparison): We finetune the pretrained PopT provided by the original authors (https://huggingface.co/PopulationTransformer) on our task data and compare that to our supervised pretrained models. We found that our pretrained models consistently outperform the pretrained PopT. This result highlights the utility of our supervised pre-training framework for extracting task-relevant representations, especially in scenarios with low quantities of task-specific data. We have added this analysis in the appendix.
>
> | Model                         | S1    | S2    | S3    |
> |------------------------------|-------|-------|-------|
> | Supervised pretrain + finetune | 0.366 | 0.321 | 0.377 |
> | Finetuned PopT               | 0.428 | 0.497 | 0.479 |
>
>
> - Additionally, we set out to evaluate the performance of PopT when pretrained on our longitudinal dataset (6.7x more data), rather than on the smaller pretraining dataset used in the original work. However, this is none trivial due to:
>     - Hyperparameter Uncertainty: The original work made specific decisions, such as using 5-second windows with a 0.2-second stride for their ensemble pre-training objective, and averaging only the center 10 time steps of the BrainBERT embeddings. Adapting these hyper-parameters is necessary for our larger dataset, but the impact of these choices on learned representations is unclear.
>     - Coordinate System: We question their choice of training on channel positions based on voxel coordinates, which are sequence-specific, instead of a standardized, participant-consistent anatomical coordinate system
>
> Given these methodological reservations and the difficulty in safely adapting hyper-parameters, we believe further experimentation on the original PopT model is needed before attempting to retrain it on our data.
>
> We wish to re-emphasize that our work addresses a fundamentally different problem than the existing SSL literature. Our main contribution is proposing a scalable training framework that leverages week-long paired audio and neural recordings, rather than focusing solely on neural recordings as is typical in SSL work. Overall, this additional analysis demonstrates that our approach outperforms a state-of-the-art sEEG decoding baseline, and highlights the importance of leveraging weeklong data whenever available.

---

> ### Author Response · Authors · 2025-11-27
> **Reply 3/3**
>
> 3. _Although the idea of using sEEG-audio signal pairs during the non-task phase to improve decoding performance during the task phase is interesting, the experimental design itself is to ensure that the subjects focus on carefully designed cognitive tasks and that the recorded sEEG signals contain information about language perception_
>
> We appreciate the reviewer’s concern regarding the neuroscientific basis of using sEEG–audio signal pairs obtained during the non-task phase. However, our approach intentionally leverages these segments to improve task-specific performance. Prior studies have shown that spontaneous or passive listening phases can still elicit reliable language-related neural activity (e.g.,[1]).
>
> Importantly, our primary goal is not to interpret the non-task signals as direct markers of cognitive processing, but rather to leverage them to scale up the available quantity of training data and improve decoding task-phase data. The observed improvement in task decoding performance suggests that these non-task sEEG–audio pairs contain consistent auditory features that improve model performance.
>
> Our preprocessing and training procedures are fully described for reproducibility and can be applied to any dataset containing time-synchronized sEEG and audio recordings, independent of task structure. This approach is grounded in the neuroscientific evidence demonstrating consistent auditory encoding across both task-engaged and non-task states.
>
> [1] Cohen, L., Salondy, P., Pallier, C., & Dehaene, S. (2021). How does inattention affect written and spoken language processing? Cortex, 138, 212–227. https://doi.org/10.1016/j.cortex.2021.02.007

---

### Official Review · Reviewer_kH4g · 2025-10-29

**Soundness:** 2
**Presentation:** 1
**Contribution:** 2
**Rating:** 2
**Confidence:** 4

**Summary:**

This paper describes week-long intracranial and audio recordings used to train a contrastive learning model. Learned representations seem to suggest that brain activity represents speech features, but that its global structure shifts, which identifies the practical problem that shift ought to be explicitly accounted-for.

**Strengths:**

- It is a strength that large amounts of data (over the course of a week) can be effectively used, apparently scalably. It is hard to assess the "over two orders of magnitude" claim (L17), though. This also reveals one of the main insights, regarding the cross-day neural drift and the need to correct for it.

**Weaknesses:**

- It is only the most minor of complaints, but the format of the Introduction is not quite typical of a scientific publication. It is suggested to omit the boldfaced headings, or to add a more narrative opening. Some claims are mentioned 'loosely' (e.g., "patients...typically spend about a week", "about 100X more neural data") or without citation. The writing generally can be tightened up and improved.
- Although references and related work are distributed throughout the paper, these tend to be isolated to specific decisions (e.g., like the wav2vec2 model used). It may have been easier to identify the apparent novelty of the work were it couched in a fulsome, contextualized background work section.
- The core of the work is a standard CLIP(-like?) contrastive alignment with typical objectives -- there's no novel architectural nor objective nor analytical
- The experiments are within-subject for a relatively small collection of patients. An ongoing problem in this community is how to either build thinker-independent models from scratch, or how to use foundation models that are generalizable, so such small-N data (in terms of patients) can be leveraged. At least for generalizability, the empirical results are narrow. Additional ablations or modifications of adjustable parameters would also be expected.

**Questions:**

- L42: Are you suggesting that there is a tradeoff between EEG and MEG in time-v-spatial resolution?
- L46: the moment participants perform an overt speech task can be disastrous for EEG. Is overt speech in EEG not included in 'typically'?
- L122: In your loss, is it the case that the objective is to pick the right V for a given U? This makes sense, but is CLIP typically symmetric?
- IBID: Negatives appear in batch only? When batch is ~128, is the number or variety of negatives modest?

---

> ### Author Response · Authors · 2025-11-27
>
> We kindly thank the reviewer for recognizing the strengths of this paper. We here provide a point-by-point response to the concerns raised.
>
> 1. _It is hard to assess the "over two orders of magnitude" claim (L17)_
>
> We refer the reviewer to Figure 1, where we state the amount of task, and non-task data. In the abstract we summarize this as from 1 hr - 100 hrs -> 2 orders of magnitude.
>
> 2. _The format of the Introduction is not quite typical of a scientific publication. It is suggested to omit the boldfaced headings, or to add a more narrative opening. Some claims are mentioned 'loosely' (e.g., "patients...typically spend about a week", "about 100X more neural data") or without citation._
>
> We thank the reviewer for the suggestion, we have removed boldfaced headings and added the following changes:
> - We added a citation for weeklong-stay for epileptic monitoring (Kim et al 2020) line 48.
> - Changed “about 100X more neural data” to “more than 100X the amount of data typically analysed in research experiments”.
>
> 3. _Although references and related work are distributed throughout the paper, these tend to be isolated to specific decisions (e.g., like the wav2vec2 model used). It may have been easier to identify the apparent novelty of the work were it couched in a fulsome, contextualized background work section._
>
> We thank the reviewer for the suggestion. We draw the reviewers attention to the paragraph starting at line 46 where we added some related but different work in neural decoding with sEEG.
>
> 4. _The core of the work is a standard CLIP(-like?) contrastive alignment with typical objectives -- there's no novel architectural nor objective nor analytical_
>
> We highlight the conceptual innovation in the scalable supervised pretraining framework using paired neural and audio data, and the new results of using out-of-distribution data via supervised pretraining to improve downstream task performance. To our knowledge, this was a first in existing literature.
>
> 5. _The experiments are within-subject for a relatively small collection of patients. An ongoing problem in this community is how to either build thinker-independent models from scratch, or how to use foundation models that are generalizable, so such small-N data (in terms of patients) can be leveraged. At least for generalizability, the empirical results are narrow. Additional ablations or modifications of adjustable parameters would also be expected._
>
> We agree that this is a core problem. sEEG research has been low on subjects and particularly low on the amount of data per subject. We crucially show with this work that the amount of valuable data per subject is far greater than previously thought. With our training paradigm, we can effectively train a model to extract much better task-relevant embeddings as evidenced by our main result (Figure 2).
>
> Questions
>
> 6. _L42: Are you suggesting that there is a tradeoff between EEG and MEG in time-v-spatial resolution?_
> We have edited this line (39) to read “however, fMRI provides limited temporal resolution, while EEG and MEG offer limited spatial resolution.”
>
> 7. _L46: the moment participants perform an overt speech task can be disastrous for EEG. Is overt speech in EEG not included in 'typically'?_
> We have edited this line to read “Overall, the brain activity used to model and decode speech perception is typically restricted to the moment where participants perform a specific cognitive task.”
>
> 8. _L122: In your loss, is it the case that the objective is to pick the right V for a given U? This makes sense, but is CLIP typically symmetric?_
> The original CLIP loss is symmetric (Radford et al, 2021). Here we use an asymmetric loss as in d’Ascoli et al., 2024, computing only the retrieval of the correct audio embedding given the brain signal, not vice-versa, since we use a frozen pretrained audio embedding model and only train the brain model.
>
> 9. _IBID (L122): Negatives appear in batch only? When batch is ~128, is the number or variety of negatives modest?_
> Retrieval set is all negatives in batch as in previous brain decoding papers using clip architecture [1]. [1] used batch size 256 which is comparable to our 128. Furthermore, this batch size allowed training on a single GPU in less than 12 hours.
>
> [1] Alexandre Defossez, Charlotte Caucheteux, Jeremy Rapin, Ori Kabeli, and Jean-Remi King. Decoding speech perception from non-invasive brain recordings. Nature Machine Intelligence, 5(10):1097–1107, October 2023. ISSN 2522-5839. Doi: 10.1038/s42256-023-00714-5.

---

### Official Review · Reviewer_YB1P · 2025-10-30

**Soundness:** 2
**Presentation:** 2
**Contribution:** 2
**Rating:** 4
**Confidence:** 4

**Summary:**

The paper introduces a supervised pretraining framework for intracranial EEG (iEEG)-based speech decoding, leveraging week-long ambient and task-based brain-audio recordings from epilepsy patients. Using a contrastive learning approach, the authors align neural signals with representations from a pretrained speech model (wav2vec 2.0), scaling dataset sizes by orders of magnitude compared to traditional short, controlled experiments. The work demonstrates that pretraining on large-scale, ambient recordings significantly improves downstream decoding performance with robust log-linear gains as data expands, while detailed representational analyses reveal substantial cross-day drift in neural embeddings.

**Strengths:**

1. Real-world relevance: The authors effectively leverage week-long clinical iEEG recordings paired with ambient audio—data typically discarded—to scale training data by over two orders of magnitude. This represents a meaningful step toward real-world, scalable brain-speech decoding and is clearly motivated and illustrated (Figure 1).

2. Rigorous and comprehensive experimental validation: The pretraining framework consistently improves downstream speech decoding across all three subjects, with statistically significant gains (Figure 2A). The log-linear scaling with pretraining data quantity (Figure 2B) and sensitivity analyses (e.g., finetuning data ablation in Figure 4A) further strengthen the claims.

3. Representational and distribution shift analysis: The paper provides a clear analysis of the distribution shift between ambient and true audiobook sounds (Figure 3) and demonstrates the necessity of finetuning. The comparison between wav2vec 2.0 and melspectrogram features (Figure 5) offers valuable insights into which acoustic representations align better with neural activity.

4. Neurophysiologically informative embedding analysis: The UMAP visualizations and linear decoding analyses (Figures 6, 10) reveal meaningful structure in the learned embeddings, particularly the day-to-day drift in neural representations—a finding with important implications for future model design and clinical translation.

**Weaknesses:**

1. Limited comparison to recent state-of-the-art baselines: The paper does not adequately situate itself within the rapidly evolving literature on neural decoding. Key recent works—such as self-supervised pretraining on iEEG [1,2] and cross-subject or cross-session transfer learning [3]—are not discussed or compared. This omission weakens the claim of methodological novelty.

2. Incomplete coverage of pretraining innovations in brain decoding: While this paper emphasizes supervised pre-training on environmental data, it lacks a detailed overview of the results from related foundational models [4,5] that also utilize large-scale neural network data. Therefore, a deeper exploration is needed regarding the connections between this work and these methods, and in what ways they represent breakthroughs.

3. Lack of neural-level interpretability and spatial ablation: The embedding analyses are informative but do not directly link to neural anatomy or functional localization. Ablations over electrode groups (e.g., auditory vs. non-auditory cortex) or analysis of how different brain regions contribute to the learned representations would strengthen the interpretability and biological plausibility of the model.

4. Superficial handling of temporal non-stationarity: Although the paper identifies day-to-day drift as a key challenge, the proposed model does not explicitly account for it. Incorporating temporal adaptation mechanisms—such as domain-adversarial training, sliding-window normalization, or time-aware embeddings—could improve robustness and generalization, and should be explored or at least discussed as a future direction.

**References:**

[1] Wu, D., Li, S., Feng, C., Cao, L., Zhang, Y., Yang, J., & Sawan, M. (2024). Towards Homogeneous Lexical Tone Decoding from Heterogeneous Intracranial Recordings. *arXiv preprint arXiv*:2410.12866.

[2] Zheng, H., Wang, H., Jiang, W., Chen, Z., He, L., Lin, P., ... & Liu, Y. (2024). Du-IN: Discrete units-guided mask modeling for decoding speech from Intracranial Neural signals. *Advances in Neural Information Processing Systems, 37*, 79996-80033.

[3] Singh, A., Thomas, T., Li, J., Hickok, G., Pitkow, X., & Tandon, N. (2025). Transfer learning via distributed brain recordings enables reliable speech decoding. *Nature Communications, 16*(1), 8749.

[4] Zhang, D., Yuan, Z., Yang, Y., Chen, J., Wang, J., & Li, Y. (2023). Brant: Foundation model for intracranial neural signal. *Advances in Neural Information Processing Systems, 36*, 26304-26321.

[5] Chau, G., Wang, C., Talukder, S., Subramaniam, V., Soedarmadji, S., Yue, Y., ... & Barbu, A. (2025). Population transformer: Learning population-level representations of neural activity. *ArXiv, arXiv*-2406.

**Questions:**

See Weaknesses.

**Details Of Ethics Concerns:**

No ethics concerns are apparent. The study protocol for data collection is IRB-approved.

---

> ### Author Response · Authors · 2025-11-27
>
> We kindly thank the reviewer for recognizing the relevance and the strengths of this work. Here we provide a point by point response to the concerns raised by the reviewer.
>
> 1. _“Limited comparison to recent state-of-the-art baselines .... Key recent works—such as self-supervised pretraining on iEEG [1,2] and cross-subject or cross-session transfer learning [3]—are not discussed or compared. This omission weakens the claim of methodological novelty.”_
>
> We thank the reviewer for the suggested references on ssl, cross-subject, and cross-session transfer learning. We have added discussion how our work relates to and is different to these work in our introduction (L53-58). We would like to highlight that the focus of this paper is not to solve the ssl or cross-subject learning problem, but rather to propose a novel framework where we demonstrate that out-of-distribution weeklong data can be leveraged to improve downstream task performance. To our knowledge, no previous paper has proposed this and shown the efficacy of this approach.
>
> 2. _Incomplete coverage of pretraining innovations in brain decoding: While this paper emphasizes supervised pre-training on environmental data, it lacks a detailed overview of the results from related foundational models [4,5] that also utilize large-scale neural network data…_
>
> We thank the reviewer for the suggested references, and would like to bring their attention to lines 436-453, where we discuss these foundation models. We also draw the reviewer’s attention to newly added lines 53-58, copied below, where we highlight how our work is different from ssl and addresses a different problem.
>
> “Current approaches to enhancing decoding performance on limited iEEG task data typically include innovations on architecture (eg. Zheng et al. (2024)), cross-subject or cross-session transfer learning (Singh et al., 2025; Wu et al., 2025; Memar et al., 2024), or self-supervised pretraining (Zhang et al., 2023; Chau et al., 2025; Yuan et al., 2024). However, there is no existing framework in which large-scale weeklong non-task neural and auditory recordings can be leveraged to improve decoding performance on downstream tasks.”
>
>
> 3. _Lack of neural-level interpretability and spatial ablation._
>
> We thank the reviewer for the suggestion and are currently running an additional analysis to address this question.
>
> 4. _Superficial handling of temporal non-stationarity_
>
> We thank the reviewer for this comment and ideas for discussion. As the focus of this paper is not handling drift, but rather the drift in the brain embedding was a discovery of learned embeddings, we have added a discussion of this as future direction in L473-478 (copied below).
>
> “Regardless of the underlying mechanism, this non-stationarity highlights the potential of designing models that are more explicitly robust under distributional drift. Methods such as domain-adversarial training (Purushotham et al., 2017), sliding-window normalization (Tanaka et al. 2022), or time-aware positional embeddings (Zhou et al., 2021) offer valuable future research directions for more drift-robust neural decoding models.”

---

> > ### Author Response · Authors · 2025-12-01
> > **Additional analysis added**
> >
> > 3. _Lack of neural-level interpretability and spatial ablation._
> >
> > We thank the reviewer for the suggestion, we have addressed this by running an analysis pretraining and finetuning per region of interest in each subject. We present the results in the appendix Figure 12. We defined three regions of interest (ROI): temporal cortex, frontal cortex and the parietal-occipital cortices. We train and test a model using only electrodes from a given ROI. For the subject (2) with a difference in performance between ROIs we observe that the temporal cortex offers a performance improvement, while the frontal and parietal-occipital result in worse performance, which fits with the established role of the temporal cortex in auditory processing [1][2].
> >
> > [1] Cathy J. Price. The anatomy of language: A review of 100 fMRI studies published in 2009.
> > Annals of the New York Academy of Sciences, 1191(1):62–88, 2010. ISSN 1749-6632. doi:
> > 10.1111/j.1749-6632.2010.05444.x
> >
> > [2] Evelina Fedorenko, Anna A. Ivanova, and Tamar I. Regev. The language network as a natu-
> > ral kind within the broader landscape of the human brain. Nature Reviews Neuroscience,
> > pp. 1–24, April 2024. ISSN 1471-0048. doi: 10.1038/s41583-024-00802-4.

---

### Official Review · Reviewer_kvB1 · 2025-10-30

**Soundness:** 2
**Presentation:** 3
**Contribution:** 3
**Rating:** 4
**Confidence:** 4

**Summary:**

The authors propose leveraging ambient audio data from long intracranial studies in a contrastive supervised pre-training stage. In turn, this enables learning from intracranial signals over the length of a study, vastly increasing the amount of training data available. The authors show that pre-training a contrastive model with this data, allows it to generalise, with some fine-tuning, to downstream speech comprehension / audio listening tasks. The results also indicate that the pre-training scales log-linearly, suggesting further data could continue to improve generalisation performance.

**Strengths:**

- Interesting idea to leverage ambient audio for a supervised pre-training stage
- Fine-tuning the pre-trained model seems to convincingly beat the baseline
- Error bars and statistical tests included show that improvements are significant
- Performs appears to scale log-linearly with pre-training data between 0-100 hours

**Weaknesses:**

- Missing baselines: Please include (1) an end-to-end baseline where you train your full architecture directly on the supervised data and (2) a baseline where you train a linear layer directly on the raw iEEG of the downstream data. Without these, it’s hard to determine whether the pre-training was necessary at all.
- Minor: Line 126-128: Özdogan et al. 2025 quotes some of the work from [A] so this should also be cited here. Similarly, line 441/442 discusses unsupervised models, for which you may also wish to cite  [B] and [C] for intracranial unsupervised foundation models.

I am open to moving towards recommending acceptance if the authors can address the above concerns satisfactorily.

[A] Jayalath, D., Landau, G. and Jones, O.P., 2025. Unlocking non-invasive brain-to-text. arXiv preprint arXiv:2505.13446.

[B] Wang, C., Subramaniam, V., Yaari, A.U., Kreiman, G., Katz, B., Cases, I. and Barbu, A., 2023. BrainBERT: Self-supervised representation learning for intracranial recordings. arXiv preprint arXiv:2302.14367.

[C] Zhang, D., Yuan, Z., Yang, Y., Chen, J., Wang, J. and Li, Y., 2023. Brant: Foundation model for intracranial neural signal. Advances in Neural Information Processing Systems, 36, pp.26304-26321.

**Questions:**

- Why resample the brain data to 40Hz for the architecture? Intracranial recordings often pick up gamma and high-gamma band frequencies that may be relevant for speech perception [D] and could improve results. The Defossez et al. (2023) architecture was designed for non-invasive (MEG) where these frequencies are often low-signal or noise, but in intracranial recordings they are likely to be useful.
- Why use the ambient data as a pre-training stage at all? What happens when you jointly train with the ambient data as well as the true audiobook data?

[D] Mugler, E.M., Patton, J.L., Flint, R.D., Wright, Z.A., Schuele, S.U., Rosenow, J., Shih, J.J., Krusienski, D.J. and Slutzky, M.W., 2014. Direct classification of all American English phonemes using signals from functional speech motor cortex. Journal of neural engineering, 11(3), p.035015.

---

> ### Author Response · Authors · 2025-12-01
>
> We kindly thank the reviewer for recognizing the relevance and the strengths of this work. Here we provide a point by point response to the concerns raised by the reviewer.
>
> 1. _Please include (1) an end-to-end baseline where you train your full architecture directly on the supervised data (2) a baseline where you train a linear layer directly on the raw iEEG of the downstream data._
>
> The end-to-end baseline training the architecture directly on supervised data is the baseline condition included in our paper (Baseline in Fig2 A). We show in our main result that when training the same architecture on task-only data, it underperforms the model with pretraining+finetuning.
>
> We thank the reviewer for the suggestion of training a linear layer directly on the raw iEEG, we have included two additional figures in the appendix (Fig. 8 and 9), where we show that our model outperforms the linear layer.
>
> 2. _Minor: Line 126-128: Özdogan et al. 2025 quotes some of the work from [A] so this should also be cited here. Similarly, line 441/442 discusses unsupervised models, for which you may also wish to cite [B] and [C] for intracranial unsupervised foundation models._
>
>
> We thank the reviewer for the suggestion and point to our citation of cite [B] in lines 433 and 442, and [C] in 433. We have added [B] to line 126 as we agree that decoding movie audio is very similar of decoding audio-book stimuli. However, we politely refrain from adding [C] to this line as the authors focus mainly on clinically-relevant downstream tasks, which is different from our cognitive decoding objective. We point the reviewer to L438 for our citation of [C] in our discussion.
>
>
> 3. _Why resample the brain data to 40Hz for the architecture? Intracranial recordings often pick up gamma and high-gamma band frequencies that may be relevant for speech perception [D] and could improve results. The Defossez et al. (2023) architecture was designed for non-invasive (MEG) where these frequencies are often low-signal or noise, but in intracranial recordings they are likely to be useful._
>
>
> We thank the review for the great suggestion - there is indeed an improvement in downstream performance using gamma band signal for 2 of the 3 subjects. We have added an additional figure (Figure 7) in the appendix showing the performance on several different neural preprocessing strategies. We are happy to update the figures in the main text to be on gamma band signal for the camera-ready version.
>
> 4. _Why use the ambient data as a pre-training stage at all? What happens when you jointly train with the ambient data as well as the true audiobook data?_
>
> We thank the reviewer for the suggestion and we have included two additional figures in the appendix (Fig. 8 and 9), where we show that adding the true audiobook data to the pretraining set makes no difference to downstream performance, it is the pretraining on ambient audio that makes a significance difference.

---

### Meta-Review · Area_Chair_rXDJ · 2025-12-25

**Summary:**

The paper introduces a framework to scale intracranial EEG (iEEG) speech decoding by leveraging week-long, continuous clinical recordings (ambient audio and neural data). The authors propose a supervised contrastive learning approach that aligns iEEG signals with wav2vec 2.0 audio representations. Key results demonstrate that pretraining on this expanded dataset improves downstream decoding performance, scaling log-linearly with data volume, and reveals significant cross-day distributional drift in neural representations.

**Reviewer Concerns:**

Addressed:
The authors effectively addressed requests for simpler and stronger baselines by including a linear probe comparison and comparing against the "PopT" foundation model (where they showed superior performance via fine-tuning).

Interpretability: The addition of an ROI-based analysis (temporal vs. frontal/parietal cortex) addressed concerns regarding the neurophysiological plausibility of the learned embeddings.

Data Clarifications: The authors clarified that while the original dataset had 46 patients, only the 3 selected had the requisite week-long recordings, justifying the sample size constraint.

Outstanding:
Limited Technical Novelty: A pervasive concern (Reviewers kH4g, awbe) is that the method relies on standard, off-the-shelf architectures (CLIP objective, wav2vec 2.0) without novel algorithmic contributions specific to the challenges of iEEG.

Small Sample Size ($N=3$): Despite the "minutes to days" scaling within-subject, the study is limited to only 3 patients. Reviewers noted this significantly hampers claims of generalizability and falls short compared to recent cross-subject or "foundation model" benchmarks in the field.

The evaluation is strictly within-subject, which reduces the impact for a venue like ICLR that increasingly values thinker-independent or few-shot generalization capabilities.

**Reviewer Scores:**

Reviewers kvB1 and YB1P (Scores: 4) acknowledged the practical value of the "ambient data" paradigm and may view the rebuttal (new baselines) positively, potentially pushing them to a borderline "Weak Accept". However, Reviewers kH4g and awbe (Scores: 2) focused on the lack of novelty and small $N$, fundamental issues that the rebuttal could not resolve. The consensus remains weighed down by the limited algorithmic contribution.

---

### Decision · Program_Chairs · 2026-01-26

Reject